# Prevalence and Associated Risk Factors of Urinary Incontinence in Sexually Active Women in Ecuador

**DOI:** 10.3390/healthcare12222296

**Published:** 2024-11-17

**Authors:** Ana Lucía Zeas-Puga, Viviana Méndez-Sacta, Bernardo Vega Crespo, Bieke Samijn, François Hervé, Patricia Martínez-Merinero, Daniel Pecos-Martín

**Affiliations:** 1Faculty of Medical Sciences, University of Cuenca, Cuenca 010204, Ecuador; 2Physiotherapy and Pain Research Centre, Department of Nursing and Physiotherapy, University of Alcalá, 28801 Madrid, Spain; 3Department of Human Structure and Repair, Ghent University, 9000 Ghent, Belgium; 4Department of Rehabilitation Sciences, Ghent University, 9000 Ghent, Belgium; 5ERN Accredited Center, Department of Urology, Ghent University Hospital, 9000 Ghent, Belgium; 6Physiotherapy and Pain Research Centre, General Foundation of the University of Alcalá, 28801 Madrid, Spain

**Keywords:** urinary incontinence, prevalence, public health, women, quality of life

## Abstract

**Background/Objectives:** Urinary incontinence (UI) significantly affects women’s health worldwide, but its specific prevalence in Ecuador is not well documented. This research aimed to determine the prevalence of urinary incontinence among sexually active women in the Cuenca canton and the factors associated with its presence. **Methods:** A descriptive cross-sectional study was conducted from August 2023 to January 2024 involving 460 women aged 30 years or older attending public health centers in Cuenca, Ecuador. The ICIQ-SF questionnaire was used to assess the presence and severity of UI and its impact on quality of life, along with a data collection form. **Results:** The findings revealed that UI is prevalent in a population that often lacks timely diagnosis, further obscuring the true extent of the issue. The results demonstrated that sociodemographic, gyneco-obstetric, and urological factors significantly influenced the risk of developing this condition. **Conclusions:** This study demonstrated a high prevalence of UI among women in Cuenca, Ecuador, with a notable impact on quality of life. The aforementioned factors predispose to the development of this condition, highlighting the need for preventive and rehabilitative interventions, as well as appropriate policies to address UI as a public health issue.

## 1. Introduction

Urinary incontinence (UI) is a highly significant public health issue that is often underestimated due to the stigma and social embarrassment surrounding it. The International Continence Society (ICS) describes it as a symptom of the storage phase and is defined as the complaint of any involuntary leakage of urine. It can be explained by various pathophysiological mechanisms, including structural and anatomical abnormalities, as well as sphincter dysfunction, among others. While UI affects both men and women, its incidence is notably higher in women [1,2]. Global prevalence estimates vary widely, with approximately 26% of women worldwide affected by UI. However, prevalence rates can range from as low as 2.5% to as high as 57.7%, highlighting the complexity and variability of this condition across different populations [3]. The prevalence of UI increases with age, as evidenced in the United Kingdom, where between six and seven million people experience some degree of UI, with approximately 39.9% of those over the age of 55–64 and 15% of those over 75–80 years old being affected. In Ecuador, studies on the prevalence of UI are limited as a result of underdiagnosis and the lack of attention given to this condition [4,5].

UI can be categorized into two groups: urgency urinary incontinence (UUI), characterized by a sudden, intense need to urinate, and stress urinary incontinence (SUI), which occurs during physical exertion or actions that increase abdominal pressure, such as sneezing. Additionally, there is mixed urinary incontinence (MUI), which combines features of both UUI and SUI. While less common, other forms of incontinence, such as overflow incontinence, which involves incomplete bladder emptying, or incontinence related to reduced bladder storage capacity, may also occur and should not be underestimated [6]. The risk factors for UI are varied, including endogenous factors such as obesity, pregnancy, and age, as well as exogenous factors like the type of childbirth, abdominal surgeries, and obstetric and perineal trauma [7]. Biomechanical alterations, such as the loss of pelvic support for the bladder and urethra, weight gain during pregnancy, shifts in the center of gravity at the sacral level, and loss of muscle tone, can also contribute to UI [8,9].

It is important to note that the relationship between UI and chronic conditions, such as cardiovascular disease and diabetes, may be bidirectional, with each condition potentially influencing the other. This complex interaction underscores the importance of a comprehensive approach to addressing both UI and related comorbidities [10]. UI significantly limits physical and social activities, leading to isolation and a diminished quality of life. Isolation can be particularly severe in older adults, where the impact of UI extends beyond the loss of bladder control. Moreover, the reduced mobility and social engagement associated with UI can increase vulnerability to a range of health issues, including cardiovascular diseases, and higher mortality rates [11].

Given the high prevalence and serious physical, psychological, and social consequences of UI, it is crucial to determine its prevalence and associated factors in the female population of Cuenca, Ecuador. This study specifically focused on sexually active women, as this group may face unique risk factors and consequences impacting quality of life and sexual health. Understanding these aspects provides valuable insights for targeted interventions within a vulnerable population. The objective of this research was to assess the prevalence of urinary incontinence among sexually active women in Cuenca Canton and the factors associated with its presence. This study aims to contribute to current knowledge, influence decision-making, guide the implementation of public policies, and serve as a foundation for future research.

## 2. Materials and Methods

We conducted a descriptive cross-sectional study between August 2023 and January 2024, aiming to evaluate the prevalence of urinary incontinence and associated risk factors among sexually active women in Cuenca Canton, Ecuador. For this study, sexually active women were defined as those who had engaged in sexual activity within the last year. Participants were recruited from health centers within Cuenca Parrish Zone 6, a region defined by the Ministry of Health of Ecuador as part of its health sectorization strategy to optimize the management of health services across different areas. The sample size was calculated based on the statistics provided by the Ministry of Health of Ecuador. The inclusion criteria focused on women aged 30 and older. Women in their 20s were excluded due to the lower reported prevalence of urinary incontinence in this age group, while older women were included to capture a broader range of experiences and potential risk factors associated with UI. A purposive sample was selected based on the records of attendance at local health centers. This study was approved by local health center authorities, and women were included after providing informed consent. Simple random sampling was employed to select a total of 460 women from health centers in the Cuenca Canton, specifically those who attended the Cuenca North and Cuenca South health centers. The inclusion criteria were being a sexually active woman aged 30 or older, having a history of urological and gynecological–obstetric conditions, and agreeing to participate in this study. Participants were excluded if they had cognitive impairments or conditions that prevented them from completing the questionnaire. Data on sociodemographic factors and associated risk factors were collected and reviewed based on similar studies. Additionally, the International Consultation on Incontinence Questionnaire—Short Form (ICIQ-SF) was used to assess the severity of UI, as it is a reliable and validated tool that highlights the prevalence, type of incontinence, severity of UI, and its correlation with quality of life [12]. This questionnaire consists of four questions. Questions 1, 2, and 4 help to determine whether urine loss is related to physical exertion or occurs suddenly in response to urgency. This questionnaire has demonstrated reliability and internal consistency, with a Kappa index ranging from 0.58 to 0.73 and Cronbach’s alpha coefficient of 0.95 [13].

### Ethical Statement

Ethical approval was granted by the Human Research Ethics Committee of the University of Cuenca, with approval code 2023-009EO-IND, and zonal coordination 6 of the Ministry of Health of Ecuador with approval code MSP-CZ6-GIR-2023-0619-M. All participants were fully briefed on this study’s purpose and provided written informed consent before the collection of samples.

For any additional inquiries regarding the data or materials used in this study, please contact this study’s authors.

## 3. Statistical Analyses

Data were analyzed using SPSS version 29.0.1.0 and Numbers 13.1. Descriptive statistics were calculated. Associations between variables were evaluated using Pearson’s chi-square test, and unadjusted odds ratios (ORs) were computed to assess the strength of these associations. A *p*-value < 0.005 was considered significant. The sample size was calculated to provide a 95% confidence level and a 5% standard error, with an estimated dropout rate of 20%.

To determine the level of risk and statistical significance, binary categorical variables were analyzed.

Participants were categorized by age into two groups: younger than 45 years and older than 45 years, with the presence or absence of UI assessed in each group. Conditions such as hysterectomy, prolapse, cardiovascular diseases, and diabetes were classified as either present or absent, enabling an analysis of their association with UI. Gynecological and obstetric history were similarly categorized, distinguishing between women who had vaginal or cesarean deliveries. Lastly, parity was classified as having one delivery or two or more, alongside the assessment of UI presence.

## 4. Results

### 4.1. Participant Characteristics

This study included a population of 460 women, with a final sample of 333 (72.4%) reporting symptoms consistent with urinary incontinence as assessed by the ICIQ-SF. The most represented age group was 30 to 49 years, with the overall age range of participants spanning from 30 to 91 years. More than half of the participants (59.8%) were engaged in the informal sector, which includes women identifying as homemakers, while those with a profession or trade fell into the formal sector. Only 17.4% of women demonstrated harmful habits, including alcoholism, smoking, poor diet, lack of adequate rest, and sedentary lifestyle. In terms of residence, the majority lived in urban areas. As for general health, cardiovascular diseases were the most prevalent condition, affecting 25.2% of participants, surpassing respiratory diseases and diabetes. Gynecological–obstetric history and other relevant health factors are further detailed in Table 1. The most prevalent type of delivery was vaginal, with 70.4% of women reporting at least one vaginal birth. Of these, 56.1% had exclusively vaginal deliveries, while the remaining had a combination of both vaginal and cesarean deliveries. The most common parity group was multiparous women (2–4 children), representing 63% of the sample.

### 4.2. Prevalence and Types of Urinary Incontinence

A total of 333 women had urinary incontinence and 127 did not, resulting in a prevalence of 72.4%. SUI was the most prevalent, affecting 220 women, which corresponds to 47.8% of the entire sample. UUI represented 3.7%, while MUI accounted for 11.1%, and other types of UI made up 9.8%.

There was variability in the severity of symptoms, with moderate UI being the most common, affecting 190 women (41.3%), followed by mild UI (21.1%), severe UI (9.4%), and very severe UI (0.7%) (Table 2). These results highlight the diversity in the presentation of UI among the participating women, both those with and without the condition.

### 4.3. Factors Associated with UI

Several demographic factors were associated with an increased risk of urinary incontinence, including age, occupation, cardiovascular diseases, and diabetes. Older women, those in informal occupational sectors, and those with certain health conditions, such as cardiovascular diseases and diabetes, showed higher odds of experiencing UI.

Additionally, reproductive history, such as parity, was found to be a significant risk factor, whereas cesarean deliveries appeared to have a protective effect. Specific details regarding each of these associations can be found in the following sections.

#### 4.3.1. Demographic and General Health-Related Factors

The prevalence of UI increased with age. In the 30- to 39-year age group, 57.2% of women had UI. This percentage rose to 72.9% in the 40- to 49-year age group and further to 80.6% in the 50- to 59-year age group. The prevalence continued to increase, reaching 84.1% in the 60- to 69-year age group and 75% in the 70- to 79-year group. In the older age groups, the prevalence was even higher, reaching 100% among women aged 80 to 89 years and those aged 90 years and older. However, it is important to acknowledge that the sample size for these older age groups was smaller compared with those of the other age ranges, which could affect the generalizability of these findings. Pearson’s chi-square test revealed a significant association between age and the prevalence of UI (OR. 2.596, 95% CI: 1.706–3.949, *p* < 0.001) (Table 3). The data also indicate that, while SUI was the most prevalent type across most age groups, UUI and MUI became more prevalent in older women.

Significant differences were also noted in UI prevalence between women working in the formal and informal sectors. A total of 216 (78.5%) women in the informal sector were affected, compared with 117 (63.2%) in the formal sector. Statistical analysis yielded an OR: 2.128, 95% CI: 1.405–3.222, *p* < 0.001, suggesting a higher prevalence of UI among informal workers, particularly homemakers (Table 3).

Among women with harmful habits, who constituted 72.5% of the sample of women with UI, no significant association with UI was found (OR: 1.007, 95% CI: 0.586–1.727, *p* = 0.4905).

Additionally, respiratory diseases showed no significant link with UI (OR: 1.207, 95% CI: 0.5298–2.752, *p* = 0.367). Conversely, there was a strong association between UI and cardiovascular diseases and diabetes, with odds ratios of 3.251 (CI: 1.807–5.848, *p* < 0.001) and 2.384 (95% CI: 1.092–5.204, *p* = 0.001), respectively. These findings indicate significant associations of cardiovascular diseases and diabetes with the presence of UI.

#### 4.3.2. Urogynaecological Factors

Parity plays a crucial role in the development of UI. In this study, we observed that women with higher parity were more likely to experience UI (*p* < 0.001, OR 2.214). Figures regarding UI and the type of delivery are summarized in Table 3. Our study found a significant relationship between UI and the type of delivery with a higher prevalence in women who had vaginal deliveries (*p* < 0.001, OR 2.899) and lower in women who had a cesarean birth history (*p* = 0.003, OR 0.5584) (Table 3).

In total, 80 women had pelvic organ prolapse. Of these, 70 (87.5%) also had UI. Statistical analysis showed a statistically significant correlation between prolapse and UI incidence (*p* < 0.001, OR 3.114).

Regarding hysterectomy, 58 women (12.6%) had undergone this procedure. Among them, 86.2% had UI, while 13.8% did not. Statistical analysis revealed a *p*-value of 0.005, which did not meet the threshold of statistical significance as defined in our methods (Table 3).

#### 4.3.3. Severity of UI and Its Impact on Quality of Life

This study also explored the relationship between the severity of UI and its impact on quality of life (QoL) in participants. Overall, 4.2% of participants with UI (n = 333) did not experience any impact on their QoL, 57.4% reported a mild impact, 23.4% experienced a moderate impact, and 15% had a severe impact.

For those with mild UI (29.1% of the women with UI), 3.6% did not perceive a significant impact on their QoL, while 25.5% experienced a low score, falling into the mild category. In the group with moderate UI (57.1%), 0.6% reported no impact, 31.5% experienced a mild impact, 19.8% a moderate impact, and 5.1% a severe impact. Among the women with severe UI (12.9%), 0.3% experienced a mild impact, 3.6% a moderate impact, and 9.0% a severe impact. For the 0.9% of women with very severe UI, all experienced a severe impact on their QoL.

Statistical analysis using Pearson’s chi-square test revealed a significant association between the severity of UI and the level of QoL impact (*p* < 0.001). Additionally, a significant association was found between the type of incontinence and the severity of UI, also with a *p*-value < 0.001.

The findings presented in Table 4 and Table 5 describe the distribution of UI types, the corresponding QoL impact, and the severity of UI among participants. The data show a significant statistical association between the severity of UI and its impact on QoL. Although there was variability in individual perceptions, the results indicate that women with more severe UI tended to report higher QoL impact. This reinforces the observed trend that increased UI severity correlates with a greater likelihood of experiencing a negative impact on QoL.

## 5. Discussion

### 5.1. General Results

Chronic health conditions, comorbidities, including urinary incontinence, and the socio-demographic structure of the aging population, invite reflection on the prevalence of pathologies that are often neglected. Although UI is not typically associated with high mortality or urgency, it is more common than previously recognized and has a significant impact on QoL. Therefore, gathering epidemiological data is essential for implementing prevention, promotion, and treatment strategies, whether medical or rehabilitative.

In the context of Ecuador, limited access to specialized medical care and cultural stigma have contributed to a scarcity of studies on UI.

In our study, the most common type of UI was stress urinary incontinence, affecting 66.1% of participants, followed by mixed urinary incontinence, at 15.3%. Regarding the impact on QoL, 57.4% of participants with UI reported a mild impact, while 23.4% experienced a moderate impact, and 15% reported a severe impact.

### 5.2. Prevalence and Type of Incontinence

Our research found that 72.4% of the participants reported some degree of UI, highlighting a considerable occurrence rate in the studied population. This aligns with findings of Milsom et al., who observed that the prevalence of UI can range from 5% and 70%, with a range of 25% to 45% for any type of UI. Prevalence rates vary between countries, as they can be influenced by cultural differences in the perception of UI and methodological differences, such as the use of different definitions [14]. Similarly, Batmani et al. reported variability in prevalence by continent, with rates of 45.1% in Asia, 46.3% in Russia, and 23% in America, noting that these differences were influenced by demographic characteristics and other associated factors, such as pregnancy, type of delivery, abdominal surgeries, and the lack of precise diagnosis [15].

Regarding the severity and types of incontinence, the most prevalent cases in our study exhibited mild to moderate severity (62.4%), which is significant given that the participants belong to a working-age, sexually active, and/or family-oriented population. Notably, as age increased, the severity also escalated, with more cases progressing to severe and very severe levels. This aligns with the findings of Rincón [16], who studied 289 women in a similar age range and reported that 79.8% of them experienced mild to moderate severity. However, our results differ from those of Alvarado [17], who observed that 70.1% of her sample of 170 women aged 20–44 presented moderate to very severe incontinence. The discrepancies may be attributed to differences in sample size, age range, and possibly methodological factors.

In our study, women aged 40 to 60 showed a predominance of 66% for SUI, making it the most common, followed by MUI, at 27%. These findings are similar to those of Todhunter et al., who observed that 50% of women under 65 years old who suffer from UI have SUI, while UUI was not common, representing 10%, and MUI showed a frequency of 30%. Likewise, Abufaraj et al., in a study conducted in the United States, found that SUI was the most common type at 45.9%, followed by UUI at 31.1% and MUI at 18.1%, with higher rates in women over 60 years old [18,19].

### 5.3. Risk Factors: General Health and Demographics

Our study demonstrated a significant association between increasing age and the prevalence of UI, consistent with global trends. UI is common in the elderly population due to age-related changes in the lower urinary tract, such as decreased bladder capacity, decreased sensation of fullness, slower detrusor muscle contraction, decreased pelvic floor muscle strength, and increased residual bladder volume. Additionally, menopause-related decreases in estrogen and collagen levels decreased the elasticity of the detrusor muscle and cause atrophic changes in the pelvic floor muscles, decreased mobility, and neurological and cognitive alterations, all of which predispose individuals to UI [15].

The informal sector, which accounted for 59.8% of the participants, particularly homemakers (47%), was the most prevalent occupation in our study. This occupation often involves repetitive and heavy-lifting activities, which predispose individuals to UI [20]. Although our study did not directly measure the impact of Valsalva maneuvers or physical strain on UI risk, the observed association between occupation and UI prevalence aligns with prior research findings. A study conducted on individuals with and without UI showed that activities involving Valsalva maneuvers and coughing led to an increase in the electromyographic activity of the pelvic floor muscles compared with those without UI. These mechanical stresses place additional pressure on the pelvic floor muscles and alter normal bladder function, which can result in pelvic floor dysfunction, contributing to the onset or worsening of UI [21,22].

We also observed a significant association between UI and chronic health conditions, particularly cardiovascular diseases and diabetes. While bi-directionality has been suggested in the literature—where each condition may potentially influence the other—this association underscores the importance of further investigation into how these interrelations might impact the QoL of individuals. Understanding how these disorders are interrelated is crucial for developing comprehensive improvement strategies. In this sense, a strong association has been observed between the risk of cardiovascular and metabolic diseases, such as diabetes and hypertension, and the presence of UI [10,23].

### 5.4. Risk Factors: Urogynaecological Factors

Gynecological and obstetric history also plays a significant role in UI. Our study observed that women who had vaginal deliveries had a 78.1% prevalence of UI, compared with 65.3% in those who had cesarean sections. This is consistent with previous studies showing that the number and type of deliveries influence continence. Youssef et al. found that the Valsalva maneuver during vaginal delivery exerts a significant impact on the pelvic floor, weakening bladder neck support and compromising its innervation [24]. With this background, vaginal delivery has a significant impact on the pelvic floor, weakening bladder neck support and compromising its innervation [25].

A cross-sectional study by Luo et al., which included 358 pregnant women, found that multiparous women experienced a higher prevalence and frequency of UI during pregnancy compared with nulliparous women. Additionally, multiparous women showed a significant descent of the bladder neck and wider urethral angles during the Valsalva maneuver. These findings coincide with the results of our research, which also revealed a high prevalence of UI among multiparous women, especially those with a history of vaginal deliveries, suggesting that the experience of multiple deliveries may be associated with an increased risk and structural changes in the pelvic floor [26].

Bozkurt et al. documented that spontaneous or surgical deliveries, perineal lacerations, and abdominopelvic surgeries can trigger pelvic floor disorders, contributing to the presentation of UI and pelvic organ prolapse [27].

In the present research, the occurrence of hysterectomy and prolapse cases was not very frequent, but it is worth noting that, among women who presented these disorders, the prevalence of UI was higher, coinciding with the findings of the aforementioned authors. This research identified that 86.2% of women who underwent a hysterectomy developed UI, coinciding with the findings of Kudish et al., who mentioned that, after this procedure, changes in the disposition of intrapelvic organs increase the risk of developing UI, especially in postmenopausal women [28,29].

Among those who experienced pelvic organ prolapse, 87.5% also reported UI. According to Tunn et al., individuals with pelvic organ prolapse may simultaneously suffer from SUI and UUI, with more severe prolapses leading to greater dysfunction and other clinical manifestations [30].

### 5.5. Impact on Quality of Life

UI impacts various aspects of life, including physical, emotional, and social spheres. Symptoms such as the urgent and frequent need to urinate and involuntary loss of urine during certain efforts can cause physical discomfort, limit social relationships and activities related to exercise, and lead to anxiety and embarrassment [31].

In the present investigation, 41.3% of the women reported a moderate impact on QoL using the ICIQ-SF. Previous studies have demonstrated that UI is significantly associated with a lower quality of life, as assessed through various health-related quality-of-life questionnaires [32]. These findings underscore the impact of UI on multiple facets of health and the benefits of targeted interventions. For instance, other studies highlight the positive impact of interventions in managing UI, where a reduction in the severity of lower urinary tract symptoms was observed, improving the quality of life; the reduction in scores was evidenced by before and after results [33].

### 5.6. Strengths, Limitations, and Recommendations

This study highlights the importance of conducting more epidemiological research on UI across diverse populations, as well as the need for education and prevention programs targeted at at-risk women. Although previous data on UI prevalence is limited, the findings of this study emphasize the significance of addressing this public health issue. This evidence serves as a foundation for developing strategies to address and support women affected by this condition, including improving access to specialized health services that offer effective treatments and rehabilitation.

One of the strengths of our study lies in the use of the ICIQ-SF questionnaire, a validated and widely recognized tool for assessing UI severity and its impact on quality of life. Given that this questionnaire has been adapted and validated in Spanish, it was highly suitable for use in our population, ensuring linguistic and cultural relevance.

However, a notable limitation of our study is the use of a non-probabilistic sampling method, which may have led to a non-homogeneous sample and limits the generalizability of our findings. Additionally, the lack of early diagnosis of UI among women and the absence of comprehensive, population-specific epidemiological data posed challenges in fully understanding the scope of the condition.

Future research should focus on addressing these limitations by using probabilistic sampling methods to improve the generalizability of findings to a broader population and by integrating early diagnostic tools. We also recommend the inclusion of non-sexually active women in future studies to provide a more comprehensive understanding of UI prevalence across various demographics. Furthermore, gathering data on the duration of UI is essential, as it will offer valuable insights into the condition’s progression and natural history, thereby supporting the development of more effective management strategies. Finally, greater attention should be paid to identifying effective preventive measures and assessing the long-term outcomes of therapeutic approaches, such as pelvic floor rehabilitation and management of comorbidities.

## 6. Conclusions

The lack of precise diagnosis and the cultural stigma associated with UI are significant barriers to adequate care and treatment.

This study reveals that the prevalence of urinary incontinence among the female population studied in Ecuador is significant, with 72.4% of participants reporting some degree of the condition. Stress urinary incontinence emerged as the most common type of UI. Various sociodemographic and health-related factors, such as age, occupation in the informal sector, cardiovascular diseases, diabetes, multiparity, and vaginal deliveries, were associated with an increased risk of UI.

These findings highlight the urgent need for targeted interventions and public health strategies to address this condition and improve the well-being of affected women.

In conclusion, this study provides valuable insights into the prevalence and risk factors of UI, laying the groundwork for future interventions and public health policies aimed at enhancing the quality of life for women dealing with this condition.

## Figures and Tables

**Table 1 healthcare-12-02296-t001:** Sociodemographic characteristics, general health conditions, and gynecological–obstetric history of the participants.

Sociodemographic Factors			
Variable	Category	n	Percentage
Age	30–39	138	30.0
	40–49	107	23.3
	50–59	93	20.2
	60–69	69	15.0
	70–79	40	8.7
	80–89	11	2.4
	90 or more	2	0.4
Occupation	Sector formal	185	40.2
	Sector Informal	275	59.8
Harmful Habits	Yes	80	17.4
	No	380	82.6
Residence	Urban	278	60.4
	Rural	182	39.6
General Health Conditions			
Cardiovascular Diseases	Yes	116	25.2
	No	344	74.8
Respiratory Diseases	Yes	33	7.2
	No	427	92.8
Diabetes	Yes	54	11.7
	No	406	88.3
Urogynecological Factors			
Hysterectomy	Yes	58	12.6
	No	402	87.0
Prolapse	Yes	80	17.4
	No	380	82.6
Gynecological—Obstetric history			
Type of Delivery	Vaginal	258	56.1
	Cesarean	107	23.3
	Both	66	14.3
	None	29	6.3
Parity	Nulliparous (0)	29	6.3
	Primiparous (1)	71	15.5
	Multiparous (2–4)	290	63.0
	Grand Multiparous (5+)	70	15.2

Percentages calculated based on the total number of study participants (n = 460).

**Table 2 healthcare-12-02296-t002:** Distribution of urinary incontinence severity and type according to the ICIQ-SF questionnaire.

Variable	Category	n	Percentage
Frequency of urine leakage	Never	127	27.6
	Once a week	153	33.2
	2–3 times a week	82	17.8
	Once a day	51	11.1
	Several times a day	31	6.7
	All the time	15	3.3
Amount of urine leakage	None	127	27.6
	A small amount	252	54.8
	A moderate amount	70	15.2
	A large amount	11	2.4
Impact on quality of life	None (0)	141	30.7
	Mild (1–3)	191	41.5
	Moderate (4–6)	78	17.0
	Severe (7–10)	50	10.9
Activity during which urine leakage occurs	Never	127	27.6
	Before reaching the toilet	85	18.5
	When coughing or sneezing	268	58.3
	While sleeping	25	5.4
	During physical activity/exercise	173	37.6
	After finishing urination and getting dressed	30	6.5
	For no obvious reason	20	4.3
	Continuously	10	2.2
Type of Incontinence	Stress Urinary Incontinence	220	47.8
	Mixed Urinary Incontinence	51	11.1
	Other types of Urinary Incontinence	45	9.8
	Urge Urinary Incontinence	17	3.7
Severity of Urinary Incontinence	Mild	97	21.1
	Moderate	190	41.3
	Severe	43	9.4
	Very Severe	3	0.7

The data in the table reflect the distribution of symptoms and severity of urinary incontinence according to the ICIQ-SF questionnaire assessment (n = 460).

**Table 3 healthcare-12-02296-t003:** Associations between clinical and sociodemographic variables and urinary incontinence.

Sociodemographic Factors							
Variable	Category	Yes UI	No UI	Total	*p*	OR	95% CI
Age	<45	124 (61.7)	77 (38.3)	201 (43.7)	<0.001	2.596	1.706–3.949
	>45	209 (80.7)	50 (19.3)	259 (56.3)			
Occupation	Formal Sector	117 (63.2)	68 (36.8)	185 (40.2)	<0.001	2.128	1.405–3.222
	Informal Sector	216 (78.5)	59 (21.5)	275 (59.8)			
Harmful Habits	Yes	58 (72.5)	22 (27.5)	80 (17.4)	0.4905	1.007	0.586–1.727
	No	275 (72.4)	105 (27.6)	380 (82.6)			
Urogynecological Factors							
Hysterectomy	Yes	50 (86.2)	8 (13.8)	58 (12.6)	0.005	2.628	1.209–5.712
	No	283 (70.4)	119 (29.6)	402 (87.4)			
Prolapse	Yes	70 (87.5)	10 (12.5)	80 (17.4)	<0.001	3.114	1.55–6.255
	No	263 (69.2)	117 (30.8)	380 (82.6)			
General Health Conditions							
Cardiovascular Diseases	Yes	101 (87.1)	15 (12.9)	116 (25.2)	<0.001	3.251	1.807–5.848
	No	232 (67.4)	112 (32.6)	344 (74.8)			
Respiratory Diseases	Yes	25 (75.8)	8 (24.2)	33 (7.2)	0.367	1.207	0.5298–2.752
	No	308 (72.1)	119 (27.9)	427 (92.8)			
Diabetes	Yes	46 (85.2)	8 (14.8)	54 (11.7)	0.001	2.384	1.092–5.204
	No	287 (70.7)	119 (29.3)	406 (88.3)			
Gynecological–Obstetric history							
Vaginal Birth History	Yes	253 (78.1)	71 (21.9)	324 (70.4)	<0.001	2.899	1.825–4.606
	No	80 (58.8)	56 (41.2)	136 (29.6)			
Cesarean Birth History	Yes	113 (65.3)	60 (34.7)	173 (37.6)	0.003	0.5584	0.3643–0.8559
	No	220 (76.7)	67 (23.3)	287 (62.4)			
Parity	Up to 1 birth	59 (59.0)	41 (41.0)	100 (21.7)	<0.001	2.214	1.389–3.530
	2 or more births	274 (76.1)	86 (23.9)	360 (78.3)			
Quality of Life							
Impact on Quality of Life	Yes	319 (100.0)	0	319 (69.3)	<0.001	10.06	6.117–16.53
	No	14 (9.9)	127 (90.1)	141 (30.7)			

The *p* values, odds ratio (OR), and 95% confidence intervals (CI) indicate the strength and significance of the association between each variable and UI. Percentages represent the proportion of women in each category affected by UI relative to the total sample in that group.

**Table 4 healthcare-12-02296-t004:** Type of incontinence, QoL impact, and urinary incontinence severity.

Variable	Category	Severity Level of Urinary Incontinence				Total	*p*
		Mild	Moderate	Severe	Very Severe		
Type of Incontinence	SUI	80 (24.0)	120 (36.0)	19 (5.7)	1 (0.3)	220 (66.1)	<0.001
	UUI	5 (1.5)	11 (3.3)	1 (0.3)	0	17 (5.1)	
	MUI	10 (3.0)	35 (10.5)	6 (1.8)	0	51 (15.3)	
	Other	2 (0.6)	24 (7.2)	17 (5.1)	2 (0.6)	45 (13.5)	
Level of Quality of Life Impact	None	12 (3.6)	2 (0.6)	0	0	14 (4.2)	<0.001
	Mild	85 (25.5)	105 (31.5)	1 (0.3)	0	191 (57.4)	
	Moderate	0	66 (19.8)	12 (3.6)	0	78 (23.4)	
	Severe	0	17 (5.1)	30 (9.0)	3 (0.9)	50 (15.0)	

Percentages are calculated based on the total number of participants with UI (n = 333).

**Table 5 healthcare-12-02296-t005:** Quality of life and type of incontinence.

Variable	Category	Type of Incontinence				Total	*p*
		SUI	UUI	MUI	Other		
Level of Quality of Life Impact	None	9 (2.7)	1 (0.3)	1 (0.3)	3 (0.9)	14 (4.2)	<0.001
	Mild	140 (42.0)	11 (3.3)	29 (8.7)	11 (3.3)	191 (57.4)	
	Moderate	46 (13.8)	3 (0.9)	16 (4.8)	13 (3.9)	78 (23.4)	
	Severe	25 (7.5)	2 (0.6)	5 (1.5)	18 (5.4)	50 (15.0)	

Percentages are calculated based on the total number of participants with UI (n = 333).

## Data Availability

The datasets generated and/or analyzed during the current study are not publicly available because they contain sensitive personal information of participants. The informed consent grants the confidentiality of the participant’s data. However, the datasets are available from the corresponding author upon reasonable request.

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
