# Peer review of "Prevalence and Associated Risk Factors of Urinary Incontinence in Sexually Active Women in Ecuador"

_healthcare, 2024, doi:10.3390/healthcare12222296_

Round 1

Reviewer 1 Report

Comments and Suggestions for Authors

Introduction
I think you need to explain why you focused on sexually active women rather than adult women.

Materials and Methods
Please describe the definition of sexually active women.
Is there a reason why you did not include people in their 20s in the age of the subjects?
On the other hand, is there a reason why you included older people in sexually active women?

Results
#1
L121: Is the “diagnosis” of UI derived from the subjective symptoms in the questionnaire (ICIQ-SF)?
It is necessary to consider whether the results of the subjective symptoms questionnaire can be called a “diagnosis”.

#2
L182: This demonstrates that cardiovascular diseases and diabetes constitute a significant risk factor for the development of UI.
I think this is more of a discussion than a result.
In the introduction, it is stated that a decline in physical ability due to urinary incontinence is a risk factor for cardiovascular disease.
Is the interpretation that cardiovascular disease and diabetes are risk factors for UI appropriate?

Discussion
#1
L280: It is stated that the factor that makes housewives more likely to develop UI is repetitive heavy work, but no evidence is given.

#2
L282: It is stated that repetitive strain and Valsalva maneuvers increase the risk of UI, but what results led to this conclusion?

#3
L290: Although it is understood from the results that there is a strong relationship between UI and cardiovascular and metabolic diseases, this needs to be revised because there is no evidence of a causal relationship or bidirectional relationship.

#4
L297: What is the relevance of this statement about obesity, chronic inflammation and UI to the results of this study?

#5
L339: ”This tool was used because it is reliable and validated, to highlight the 339 prevalence, type of incontinence, and severity of UI, and its correlation with quality of life 340 [29,30].”  Shouldn't this be written in the Methods section?

#6
L344: The fact that the SF-36 form was used is not mentioned in the Methods section. It also is not clearly stated in the Results section, so it needs to be added.

Author Response

Dear Reviewer,

We sincerely appreciate your thoughtful feedback and valuable insights, which have significantly improved the quality of our manuscript. We have carefully considered each of your comments and revised the manuscript accordingly.

Below, we provide a point-by-point response to each comment, with references to the specific sections and line numbers where changes were made.

Comment:

Introduction
I think you need to explain why you focused on sexually active women rather than adult women.

Thank you for your comment. We have added a sentence in the introduction (L70) to clarify our focus on sexually active women, explaining that this group may face unique risk factors and impacts on quality of life and sexual health.

Comment:

Materials and Methods
Please describe the definition of sexually active women.
Is there a reason why you did not include people in their 20s in the age of the subjects?
On the other hand, is there a reason why you included older people in sexually active women?

Thank you for your valuable comment. We have added the definition of sexually of sexually active women in the Materials and Methods section (L81), stating that they were defined as those who had engaged in sexual activity within the last year. Regarding the age group, women in their 20s were excluded due to the lower reported prevalence of urinary incontinence in this age group. On the other hand, older women were included to capture a broader range of experiences and potential risk factors associated with UI (L86).

Comment:

Results

#1

L121: Is the “diagnosis” of UI derived from the subjective symptoms in the questionnaire (ICIQ-SF)?
It is necessary to consider whether the results of the subjective symptoms questionnaire can be called a “diagnosis”.

Thank you for your thoughtful comment. We have clarified that the diagnosis of urinary incontinence was not made based solely on the subjective symptoms reported in the ICIQ-SF questionnaire. Instead, the study included a population of 460 women, with a final sample of 333 (72.4%) reporting symptoms consistent with urinary incontinence as assessed by the ICIQ-SF (L134).

We acknowledge that the questionnaire assesses symptoms rather than providing a formal clinical diagnosis.

#2
L182: This demonstrates that cardiovascular diseases and diabetes constitute a significant risk factor for the development of UI.
I think this is more of a discussion than a result.
In the introduction, it is stated that a decline in physical ability due to urinary incontinence is a risk factor for cardiovascular disease.
Is the interpretation that cardiovascular disease and diabetes are risk factors for UI appropriate?

Thank you for your valuable feedback. In response to your comment, we have revised the wording in the results section. We now state that “These findings indicate significant associations of cardiovascular diseases and diabetes with the presence of UI” (L195).

Additionally, the introduction has been updated to clarify that the relationship between UI and chronic conditions like cardiovascular disease and diabetes may be bidirectional, with each influencing the other (L59).

Comment:

Discussion

#1
L280: It is stated that the factor that makes housewives more likely to develop UI is repetitive heavy work, but no evidence is given.

Thank you for your comment. We acknowledge that our study did not directly evaluate the effects of Valsalva maneuvers or repetitive strain. However, we have revised the text to clarify that the association between this occupation and UI is supported by previous studies, as referenced (L295): “This occupation often involves repetitive and heavy-lifting activities, which predispose individuals to UI [20].”

#2
L282: It is stated that repetitive strain and Valsalva maneuvers increase the risk of UI, but what results led to this conclusion?

Thank you for your comment. We acknowledge that our study did not directly assess the impact of Valsalva maneuvers or physical strain on the risk of UI. However, we have revised the manuscript to clarify that the association between occupation and UI prevalence is consistent with findings from prior research. We have added the following statement: “Although our study did not directly measure the impact of Valsalva maneuvers or physical strain on UI risk, the observed association between occupation and UI prevalence aligns with prior research findings” (L296).

#3
L290: Although it is understood from the results that there is a strong relationship between UI and cardiovascular and metabolic diseases, this needs to be revised because there is no evidence of a causal relationship or bidirectional relationship.

Thank you for your comment. We have revised the text to clarify that although a bidirectional relationship has been suggested in the literature, it has not been definitively established. We now state: We also observed a significant association between UI and chronic health conditions, particularly cardiovascular diseases and diabetes. While bi-directionality has been suggested in the literature, further research is needed to understand these interrelations and their impact on quality of life. (L304)

#4
L297: What is the relevance of this statement about obesity, chronic inflammation, and UI to the results of this study?

Thank you for your comment. We agree that this statement was not directly relevant to the results of our study. As such, we have removed this section, along with the reference to bibliographic source [21], to maintain focus on the findings of our study (L311).

#5
L339: ”This tool was used because it is reliable and validated, to highlight the 339 prevalence, type of incontinence, and severity of UI, and its correlation with quality of life 340 [29,30].”
  Shouldn't this be written in the Methods section?

Thank you for your valuable suggestion. We agree that this information is more appropriate for the Methods section. We have moved this explanation to the Methods section and revised it accordingly: “Additionally, the International Consultation on Incontinence Questionnaire – Short Form (ICIQ-SF) was used to assess the severity of UI, as it is a reliable and validated tool that highlights the prevalence, type of incontinence, severity of UI, and its correlation with quality of life.” (L101)

#6
L344: The fact that the SF-36 form was used is not mentioned in the Methods section. It also is not clearly stated in the Results section, so it needs to be added.

Thank you for pointing this out. We apologize for the confusion. To clarify, although we referenced studies that have used the SF-36 to assess quality of life in the Discussion section, we did not use this specific questionnaire in our study. The mention of SF-36 was intended to introduce findings from other research regarding the relationship between urinary incontinence and quality of life. We have updated the text to reflect this clarification: “Previous studies have demonstrated that UI is significantly associated with a lower quality of life, as assessed through various health-related quality-of-life questionnaires.” (L349).

Once again, we deeply appreciate your constructive feedback, which has contributed to enhancing the clarity and quality of our manuscript.

We hope the revisions meet your expectations, and we look forward to your further comments.

Reviewer 2 Report

Comments and Suggestions for Authors

The article titled "Prevalence and Associated Risk Factors of Urinary Incontinence in Sexually Active Women in Ecuador" provides valuable insights into the prevalence and risk factors of urinary incontinence (UI) among sexually active women in Cuenca, Ecuador.   

The study found that UI is prevalent in this population, with stress urinary incontinence (SUI) being the most common type. The research also identified various sociodemographic and health-related factors associated with an increased risk of UI, including age, occupation in the informal sector, cardiovascular diseases, diabetes, multiparity, and vaginal deliveries.   

However, there are some areas where the study could be improved:

  • Sampling method: The study used a non-probabilistic sampling method, which may limit the generalizability of the findings.
  • Population: The study only included sexually active women, which may not accurately reflect the prevalence of UI in the general population.
  • Data collection: The study did not collect data on the duration of UI, which may limit the ability to draw conclusions about the natural history of the condition.

To enhance the section on recommendations for future research, consider adding the following points:

  • Probabilistic Sampling: Suggest employing probabilistic sampling methods in future studies to improve the generalizability of findings to a broader population.

  • Inclusion of Non-sexually Active Women: Recommend incorporating non-sexually active women in future research to gain a more comprehensive understanding of UI prevalence across various demographics.   

  • Duration of UI: Emphasize the importance of gathering data on the duration of UI in future studies. This will provide valuable insights into the progression and natural history of the condition, facilitating better management strategies.

Despite these limitations, the study provides valuable insights into the prevalence and risk factors of UI in sexually active women in Cuenca, Ecuador. The findings of this study can be used to develop targeted interventions and public health strategies to address this condition.   

Author Response

Dear Reviewer,

We sincerely appreciate your thoughtful feedback and valuable insights, which have significantly improved the quality of our manuscript. We have carefully considered each of your comments and revised the manuscript accordingly.

To help facilitate your review, we have highlighted all the changes made in response to your comments in blue within the manuscript.

Below, we provide a point-by-point response to each comment, with references to the specific sections and line numbers where changes were made.

Comment:

Sampling method: The study used a non-probabilistic sampling method, which may limit the generalizability of the findings.

Thank you for your comment. We acknowledge that the use of a non-probabilistic sampling method may limit the generalizability of our findings. However, considering that this is the first study conducted in the city of Cuenca, Ecuador, we believe that the data gathered still provide important insights that could be valuable for future research and serve as a useful reference for subsequent studies.

Comment:

Population: The study only included sexually active women, which may not accurately reflect the prevalence of UI in the general population. 

Thank you for your valuable comment. We recognized that the inclusion of only sexually active women may limit the generalizability of our findings to the broader population. However, we selected this specific group because they may experience unique risk factors and challenges related to urinary incontinence and sexual health.

Comment:

Data collection: The study did not collect data on the duration of UI, which may limit the ability to draw conclusions about the natural history of the condition.

Thank you for your thoughtful comment. We agree that the lack of data on the duration of urinary incontinence is a limitation of our study. The duration of UI is an important factor that could provide valuable insights into the natural history of the condition. While this information was not collected in our study, we recognize its importance and suggest that future research should address this gap to further enhance our understanding of the progression of UI.

Comment:

To enhance the section on recommendations for future research, consider adding the following points:
Probabilistic Sampling: Suggest employing probabilistic sampling methods in future studies to improve the generalizability of findings to a broader population.

Thank you for your suggestion. We have acknowledged this limitation in our study and agree that employing probabilistic sampling methods in future research could enhance the generalizability of findings to a broader population. We have included this recommendation in the “Strengths, Limitations, and Recommendations” section (L373).

Inclusion of Non-sexually Active Women: Recommend incorporating non-sexually active women in future research to gain a more comprehensive understanding of UI prevalence across various demographics.  

We appreciate your valuable suggestion. We agree that including non-sexually active women in future studies would provide a more comprehensive understanding of UI prevalence across diverse demographics. This recommendation has been added to the “Strengths, Limitations, and Recommendations” section (L375).

Duration of UI: Emphasize the importance of gathering data on the duration of UI in future studies. This will provide valuable insights into the progression and natural history of the condition, facilitating better management strategies.

Thank you for highlighting this important point. We agree that collecting data on the duration of UI is crucial for understanding its progression and natural history, which would enhance the development of more effective management strategies. We have incorporated this into the “Strengths, Limitations, and Recommendations” section (L377).

Once again, we deeply appreciate your constructive feedback, which has contributed to enhancing the clarity and quality of our manuscript.

We hope the revisions meet your expectations, and we look forward to your further comments.